# Electrically Induced Sensory Trick in a Patient with Musician’s Dystonia: A Case Report

**DOI:** 10.3390/brainsci13020223

**Published:** 2023-01-29

**Authors:** Daisuke Nishida, Katsuhiro Mizuno, Osamu Takahashi, Meigen Liu, Tetsuya Tsuji

**Affiliations:** 1Department of Rehabilitation Medicine, Tokai University School of Medicine, Kanagawa 259-1193, Japan; 2Department of Rehabilitation Medicine, Keio University School of Medicine, Tokyo 160-8582, Japan

**Keywords:** musician’s dystonia, electrical stimulation, sensory trick, surface electromyograms

## Abstract

A sensory trick is a specific maneuver that temporarily improves focal dystonia. We describe a case of musician’s dystonia in the right-hand fingers of a patient, who showed good and immediate improvement after using an electrical stimulation-mimicking sensory trick. A 49-year-old professional guitarist presented with chronic involuntary flexion of the right-hand third and fourth fingers that occurred during guitar performances. Electrical stimulation with a frequency of 40 Hz and an intensity of 1.5 times the sensory threshold was administered on the third and fourth fingernails of the right hand, which facilitated fluent guitar playing. While he played guitar with and without electrical stimulation, we measured the surface electromyograms (sEMG) of the right extensor digitorum and flexor digitorum superficialis muscles to evaluate the sensory-trick-like effects of electrical stimulation. This phenomenon can offer clues for developing electrical stimulation-based treatment devices for focal dystonia. Electrical stimulation has the advantage that it can be turned off to avoid habituation. Moreover, the device is easy to use and portable. These findings warrant further investigation into the use of sensory stimulation for treating focal dystonia.

## 1. Introduction

A sensory trick is a specific maneuver that temporarily improves focal dystonia, and this effect is often observed in clinical settings. Most effects are provoked by internal stimulation (e.g., touching the skin by oneself [1]), whereas a few are provoked by external stimulation (e.g., goggles for blepharospasm [2] and hanger for cervical dystonia [3]). The sensory trick has an effect rate of 44–82% in cranio-cervical dystonia and 20–38% in upper limb dystonia [4]. When task-specific and non-task-specific dystonia are compared, task-specific dystonia presents in 38.5% of cases, and the peak age of dystonia onset is in the fifties for non-task-specific dystonia, whereas it is in the thirties for task-specific dystonia. A previous study reported that there were no significant differences between the two dystonia groups in terms of sensory trick frequency [5].

Task-specific focal dystonia in musicians, which is called musician’s dystonia [6], occurs frequently in a similar fashion to other forms of occupation-related dystonia. These are usually caused by repeated activities, such as handwriting, computer keyboard typing, or performing surgery [7,8]. Previous studies have shown the distribution of dystonic hand patterns and musical instrument class use, where finger flexion was the most prevalent dystonic movement in instrument players. It occurs in 25–76% of musician’s dystonia cases [9]. Plucked string instrument players, including guitar players, are the second most common group of players to suffer from dystonia, only after keyboard instrumentalists [7,9].

However, to date, there is limited knowledge regarding the phenomenology, pathophysiology, and effective treatment for musician’s dystonia. Herein, we describe the case of a professional guitarist who developed musician’s dystonia in his right-hand fingers but showed good and immediate improvement after using an electrical stimulation-mimicking sensory trick.

## 2. Case Presentation

A 49-year-old professional guitarist presented with a 7-year history of involuntary flexion of the right-hand third and fourth fingers while playing the guitar. He had studied guitar since the age of 12 years and started working as a professional guitar player at the age of 24 years in Germany. The symptoms emerged a few months after his last guitar playing competition. Initially, he noticed a fine involuntary muscle contraction in the affected fingers when performing rapid and repetitive movements. Gradually, the involuntary muscle contractions increased during performances and daily living activities. Previously, he was treated with oral medications (i.e., clonazepam, etizolam, mexiletine, and Syakuyaku-kanzoto extract [Kampo medicine]), which did not prove to be effective. The neurological examination findings were normal, except for the involuntary muscle contractions in the right-hand third and fourth digits that occurred while playing the guitar or when extending the fingers before grasping objects. His grip strength was 28 kg/25 kg, and the manual muscle test score of the extensor digitorum and flexor digitorum superficialis was 5/5 (right/left, respectively). When he tried to extend his right fingers, not only the finger extensor muscles (e.g., extensor digitorum communis) but also the flexor muscles (e.g., flexor digitorum superficialis and profundus) contracted. Neural imaging studies, including computed tomography and magnetic resonance imaging, indicated no abnormalities. Thus, we diagnosed it as idiopathic task-specific dystonia, according to the Japanese clinical guidelines [10,11]. His symptoms temporarily improved if either of the fingers were slightly touched by himself or with objects, such as tables. This phenomenon can be described as a “sensory trick” with external stimulation. Therefore, we tried to apply this sensory trick as a treatment using electrical stimulation.

To explore the most effective stimulus frequency, we tested electrical stimulations of 20, 40, 50, 70, and 100 Hz. As a result, we found that stimulation with a frequency of 40 Hz and an intensity of 1.5 times the sensory threshold administered on the third and fourth fingernails of the right hand was the most effective in facilitating fluent guitar playing. We measured the surface electromyograms (sEMG) of the right extensor digitorum and right flexor digitorum superficialis muscles while he was playing the guitar (1) without (pre), (2) with, and (3) without (post) electrical stimulation. The sEMG findings showed strong flexor contraction without stimulation (pre), reduced contraction with electrical stimulation, and re-increased contraction without stimulation (post) (Figure 1). He could play the guitar at almost proper tempo with electrical stimulation, whereas the tempo became incorrect when he played without the electrical stimulation (Appendix A). However, this sensory trick using electrical stimulation had not been used in everyday life to date. He practiced playing the guitar with his dystonic fingers using the effect of the sensory trick by himself and gradually improved; unfortunately, he could not achieve complete recovery of function. Thus, he was not able to return as a professional player and was teaching at his own guitar school.

## 3. Discussion

Although the sensory trick is often an effective treatment for dystonia, the mechanisms of the sensory trick are still unknown. Some studies have shown the activation patterns of the brain using neuroimaging approaches. Nauman et al. [12] investigated the brain activation patterns of patients with cervical dystonia using positron emission tomography. They showed that sensory tricks may induce a rebalancing of the central processing by reducing the activation of the supplementary motor area and primary sensory motor cortex. Cho et al. [13] showed that the connectivity of the supplementary motor area increased in the brain regions involved in sensorimotor integration during a sensory trick. These studies have suggested that sensory tricks utilize cortical activation and the network involved in motor preparation and sensorimotor integration. Regarding the electrical stimulation as a “sensory trick,” some studies [14,15,16] have investigated the brain activation patterns of peripheral electrical stimulation with somatosensory-evoked potentials. In those studies, the patients with cervical dystonia had an abnormal response to 1 or 20-Hz repetitive sensory stimulation and had abnormally sensitive homeostatic mechanisms of inhibitory circuitry in both the sensory and motor systems. Those studies focused on the possibility to modulate dystonia by prolonged electrical stimulation, in an attempt to induce plasticity. However, our study findings suggested an immediate effect during the stimulation owing to a mechanism that has not yet been defined. To speculate the immediate effect of electrical stimulation, the mechanisms that underlie sensory information processing and sensorimotor integration in dystonia might help understand this phenomenon. For instance, it was found that patients with writer’s cramp lacked the normal gating of afferent information because of a disordered central control of afferent inputs [17,18]. Moreover, an abnormal somatotopic digital representation in the primary somatosensory cortex (S1) has been demonstrated in focal hand dystonia cases [19]. Unfortunately, we did not investigate the brain physiological functions of this patient.

To understand the brain activity underlying musician’s dystonia, Furuya et al. [8] investigated the brain activity and dystonic movements of patients with musician’s dystonia. By measuring the motor evoked potential, they showed that reduced inhibition at the motor cortex was related to the temporal imprecision of skilled finger movements. Furthermore, an elevated facilitation of the motor cortex was associated with an abnormally sluggish transition of the finger movements from flexion to extension, which was the same as our case. In a study using functional magnetic resonance imaging, Pujol et al. [20] showed a significantly larger activation of the contralateral primary sensorimotor cortex that contrasted with a conspicuous bilateral underactivation of the premotor areas in guitarists with musician’s dystonia [9]. In addition, Murase et al. [19] investigated the attenuation of somatosensory evoked potentials for the sensorimotor link in patients with writer’s dystonia. They showed that both active and passive movements induced a sensory gating effect in patients with focal hand dystonia. These studies suggest that increased excitability in the sensorimotor areas plays an important role in the emergence of focal dystonias, including musician’s dystonia, while a sensory trick as sensory input corrects the imbalance of the sensorimotor integration.

Our report suggests that electrical stimulation as an external stimulus can be effective at a proper frequency customized for each patient. Notably, a previous study showed that a 10-Hz electrical stimulation could transiently improve cervical dystonia [21]. Several studies have used neuromuscular electrical stimulation with frequencies between 10 and 100 Hz for rehabilitation training in order to enhance afferent-sensory inputs and motor activation [22]. Therefore, electrical stimuli with frequencies in this range, as used in our case, likely modulate sensory input from dystonic muscles [23]. Continuous external stimulation with orthotics or other devices causes habituation and the effect often wears off within a short period. The advantages of electrical stimulation are that it can be turned off to avoid habituation, in addition to the device being simple and portable. Therefore, our case warrants further investigation into the use of sensory stimulation for treating focal dystonia.

## 4. Conclusions

In this case report, electrical stimulation was effective as a “sensory trick” in treating focal dystonia. Our results suggest that applying a sensory trick, which is usually considered to be caused by an active process (such as touching oneself), can be induced by external sensory stimulation in some cases. This phenomenon can offer clues for developing electrical stimulation-based treatment devices for focal dystonia.

## Figures and Tables

**Figure 1 brainsci-13-00223-f001:**
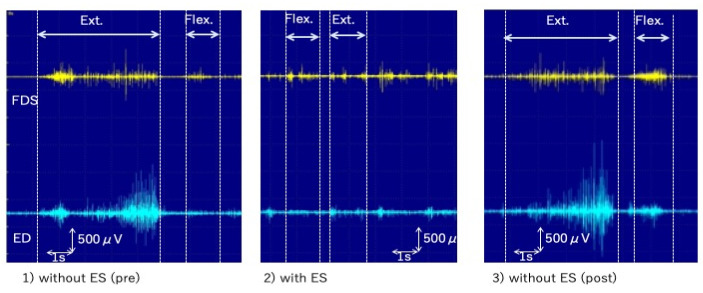
sEMG with or without the electrical stimulation. ES: the electrical stimulation, Ext.: extension, Flex.: Flexion, ED: extensor digitorum muscle, FDS: flexor digitorum superficialis muscle, sEMG: surface electromyogram.

## Data Availability

The data that support the findings of this study are available from the corresponding author upon reasonable request.

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
