# Peer review of "Electrically Induced Sensory Trick in a Patient with Musician’s Dystonia: A Case Report"

_brainsci, 2023, doi:10.3390/brainsci13020223_

Round 1
Reviewer 1 Report
The authors reported a case of electrically induced sensory trick in a patient with musician’s dystonia. I have some comments to the authors:
- Please ask to a native English speaker to check the language.
- After the first line the authors should include the frequency of sensory trick in cranio-cervical dystonia (44-82%) and the overall occurrence in upper limb dystonia (20-38%). Therefore, this study should be included in the references’ list:
Dagostino S, et al. Sensory trick in upper limb dystonia. Parkinsonism Relat Disord. 2019 Jun;63:221-223.
- Please also refer to general information of non-task specific upper limb dystonia, which is a frequent condition, often neglected. This recent big study summarized the differences between the two conditions and it should be included in the text:
Defazio G, et al. Idiopathic Non-task-Specific Upper Limb Dystonia, a Neglected Form of Dystonia. Mov Disord. 2020 Nov;35(11):2038-2045.
- “Previously, he was treated with oral medications, which did not prove effective.” Please specify which kind of medications and the rationale.
- How did the authors perform the diagnosis? Please specify the recommendation/guidelines that they used.
- Did the patient perceive the amelioration? Please specify
Author Response
Reviewer1
Thank you for your educational comments.
The authors reported a case of electrically induced sensory trick in a patient with musician’s dystonia. I have some comments to the authors:
- Please ask to a native English speaker to check the language.
Response: Thank you for your educational comments. The first manuscript and this revised manuscript has been checked by Editage (www.editage.jp) for English language.
- After the first line the authors should include the frequency of sensory trick in cranio-cervical dystonia (44-82%) and the overall occurrence in upper limb dystonia (20-38%). Therefore, this study should be included in the references’ list:
Dagostino S, et al. Sensory trick in upper limb dystonia. Parkinsonism Relat Disord. 2019 Jun;63:221-223.
A: We had read this paper prior to writing our manuscript but failed to state the frequency. We have added this detail to the manuscript (Reference no. 4).
- Please also refer to general information of non-task specific upper limb dystonia, which is a frequent condition, often neglected. This recent big study summarized the differences between the two conditions and it should be included in the text:
Defazio G, et al. Idiopathic Non-task-Specific Upper Limb Dystonia, a Neglected Form of Dystonia. Mov Disord. 2020 Nov;35(11):2038-2045. 2
A: We had failed to include information regarding non-task-specific upper limb dystonia. We have added this information in the introduction (Reference no.5).
- “Previously, he was treated with oral medications, which did not prove effective.” Please specify which kind of medications and the rationale.
A: Thank you for the comment. We have added the names of the medicines. (lines 57–58)
- How did the authors perform the diagnosis? Please specify the recommendation/guidelines that they used.
A: We have added the clinical guidelines and the diagnostic process for this patient.(Reference no.10, 11; line no.68)
- Did the patient perceive the amelioration? Please specify
A: He tried to play the guitar with his dystonic fingers using the effect of the sensory trick by himself and gradually improved; unfortunately, he could not achieve complete recovery of function. Thus, he was not able to return as a professional player and was teaching at his own guitar school.
Reviewer 2 Report
Thank you for the opportunity to review this manuscript. The authors have presented a very interesting case of improvement in task specific dystonia with electrical stimulation. This intervention can potentially serve as a guide for therapies for such dystonia. Overall, this is a well written paper. However, I would recommend that the authors provide a better description of patient's dystonia in neurological exam section (lines 52-54), , detailing the muscles involved and their respective actions.
Author Response
Thank you for comment.
Thank you for the opportunity to review this manuscript. The authors have presented a very interesting case of improvement in task specific dystonia with electrical stimulation. This intervention can potentially serve as a guide for therapies for such dystonia. Overall, this is a well written paper. However, I would recommend that the authors provide a better description of patient's dystonia in neurological exam section (lines 61-66), , detailing the muscles involved and their respective actions.
A: Thank you for the comment. We have added the description of the patient’s dystonia in detail (lines 67–72).
Reviewer 3 Report
The current study was a single subject case study in a person with focal dystonia (49 year old professional guitarist) in the fingers of the right hand. The overall purpose of the current study was to examine the influence of the use of electrical stimulation as a sensory trick to improve motor symptoms. Traditionally, it has been shown that various sensory tricks (light touches to an area near the dystonia) can transiently improve motor symptoms. Thus, this study wanted to determine if electrical stimulation could be a viable alternative to traditional sensory tricks as there could be advantages to doing it electrically.
The patient ultimately received electrical stimulation (40 Hz, 1.5x sensory threshold) to the third and fourth fingers of the right hand during guitar playing while EMG was measured from the extensor digitorum and flexor digitorum muscles. The EMG results showed reduced flexor EMG during stimulation which was the main finding.
In the discussion the authors concluded: 1) electrical stimulation as an external stimulus can be effective in focal dystonia if the frequency in individualized for a given person as more than just the 40 Hz frequency was applied, but 40 worked best for this patient; and 2) electrical stimulation using the device and methodology in the current study has the advantages that it can be turned off to avoid habituation, it is simple to use, and portable. I would also assume it is low cost,
Overall, I liked this manuscript and I think it should be published once a few weaknesses are addressed (see below). I feel it could add significantly to the literature and the electrical stimulation methodology used here could eventually be used in several research and rehabilitation settings for focal hand dystonia.
The strengths of the study were the somewhat novel design and research questions along with the concurrent EMG data. In addition, the study has a good deal of potential practical significance.
I don’t think there were any major weaknesses or fatal flaws. The greatest weakness was the typical weaknesses associated with case studies, which is obviously to be expected.
Other minor weaknesses the authors should address include:
1. Were there any guitar playing objective or subjective measures of performance taken? Or was the only dependent variable the EMG. The EMG is fine but one would like to see any associated performance changes if they could be measured. Was anything like this done? This information should be included or addressed even if it were not conducted. Please add this as appropriate.
2. I think the first paragraph of the Discussion needs some improvement. A little historical background information of sensory tricks in dystonia and overall findings should be given. As it stands now the authors seem to have picked several different unrelated studies and just listed them one by one without really linking them all together in a coherent way. I would suggest trying to improve this section and focusing on the most recent research that has provided some insights into the physiology underlying sensory tricks.
Minor typos etc.
3. Line 44, put a comma before the word “but”
4. Line 48 should probably be reworded to something like “while playing the guitar”
5. Line 90 the reference number nine should not be superscript
6. Line 91 the word writer’s has too many spaces after the apostrophe
Author Response
Thank you for your comments.
The current study was a single subject case study in a person with focal dystonia (49 year old professional guitarist) in the fingers of the right hand. The overall purpose of the current study was to examine the influence of the use of electrical stimulation as a sensory trick to improve motor symptoms. Traditionally, it has been shown that various sensory tricks (light touches to an area near the dystonia) can transiently improve motor symptoms. Thus, this study wanted to determine if electrical stimulation could be a viable alternative to traditional sensory tricks as there could be advantages to doing it electrically.
The patient ultimately received electrical stimulation (40 Hz, 1.5x sensory threshold) to the third and fourth fingers of the right hand during guitar playing while EMG was measured from the extensor digitorum and flexor digitorum muscles. The EMG results showed reduced flexor EMG during stimulation which was the main finding.
In the discussion the authors concluded: 1) electrical stimulation as an external stimulus can be effective in focal dystonia if the frequency in individualized for a given person as more than just the 40 Hz frequency was applied, but 40 worked best for this patient; and 2) electrical stimulation using the device and methodology in the current study has the advantages that it can be turned off to avoid habituation, it is simple to use, and portable. I would also assume it is low cost.
Overall, I liked this manuscript and I think it should be published once a few weaknesses are addressed (see below). I feel it could add significantly to the literature and the electrical stimulation methodology used here could eventually be used in several research and rehabilitation settings for focal hand dystonia.
The strengths of the study were the somewhat novel design and research questions along with the concurrent EMG data. In addition, the study has a good deal of potential practical significance.
I don’t think there were any major weaknesses or fatal flaws. The greatest weakness was the typical weaknesses associated with case studies, which is obviously to be expected.
Other minor weaknesses the authors should address include:
- Were there any guitar playing objective or subjective measures of performance taken? Or was the only dependent variable the EMG. The EMG is fine but one would like to see any associated performance changes if they could be measured. Was anything like this done? This information should be included or addressed even if it were not conducted. Please add this as appropriate.
A: Thank you for your kind comments. I have revised to describe the sound of the guitar while playing. Unfortunately, it is difficult to analyze the dispersion of rhythm so I just included the video clip (Supplementary Video 1) and referred the readers to watch and listen to the video clip of the guitar being played with or without the electrical stimulation.
- I think the first paragraph of the Discussion needs some improvement. A little historical background information of sensory tricks in dystonia and overall findings should be given. As it stands now the authors seem to have picked several different unrelated studies and just listed them one by one without really linking them all together in a coherent way. I would suggest trying to improve this section and focusing on the most recent research that has provided some insights into the physiology underlying sensory tricks.
A: Thank you for the comment. I have revised the discussion to focus on insights into the physiology on sensory tricks (lines 93–102)
Minor typos etc.
- Line 44, put a comma before the word “but”
A: I corrected this point.
- Line 48 should probably be reworded to something like “while playing the guitar”
A: I revised the sentence (line 53).
- Line 90 the reference number nine should not be superscript
- Line 91 the word writer’s has too many spaces after the apostrophe
A: I made corrections for points 5 and 6.
Round 2
Reviewer 1 Report
The authors have addressed all the points.
Author Response
Thank you very much for review and commented 'The authors have addressed all the points.'